# Investigating Bioactive-Glass-Infused Gels for Enamel Remineralization: An In Vitro Study

**DOI:** 10.3390/jfb15050119

**Published:** 2024-04-29

**Authors:** Zbigniew Raszewski, Katarzyna Chojnacka, Marcin Mikulewicz

**Affiliations:** 1SpofaDental, Markova 238, 506-01 Jicin, Czech Republic; 2Department of Advanced Material Technologies, Faculty of Chemistry, Wroclaw University of Science and Technology, Smoluchowskiego 25, 50-372 Wroclaw, Poland; katarzyna.chojnacka@pwr.edu.pl; 3Department of Dentofacial Orthopaedics and Orthodontics, Division of Facial Abnormalities, Wroclaw Medical University, Krakowska 26, 50-425 Wroclaw, Poland; marcin.mikulewicz@umw.edu.pl

**Keywords:** bioactive glass, dental gel, remineralization, ion release, enamel repair

## Abstract

Objective: Dental hypersensitivity remains widespread, underscoring the need for materials that can effectively seal dental tubules. This study evaluated the potential of bioactive-glass-infused hydroxyethyl cellulose gels in this context. Methods: Five gels were synthesized, each containing 20% bioactive glass (specifically, 45S5, S53P4, Biomin F, and Biomin C), with an additional blank gel serving as a control. Subjected to two months of accelerated aging at 37 ± 2 °C, these gels were assessed for key properties: viscosity, water disintegration time, pH level, consistency, adhesion to glass, and element release capability. Results: Across the board, the gels facilitated the release of calcium, phosphate, and silicon ions, raising the pH from 9.00 ± 0.10 to 9.7 ± 0.0—a range conducive to remineralization. Dissolution in water occurred within 30–50 min post-application. Viscosity readings showed variability, with 45S5 reaching 6337 ± 24 mPa/s and Biomin F at 3269 ± 18 mPa/s after two months. Initial adhesion for the blank gel was measured at 0.27 ± 0.04 Pa, increasing to 0.73 ± 0.06 Pa for the others over time. Gels can release elements upon contact with water (Ca^−^ Biomin C 104.8 ± 15.7 mg/L; Na^−^ Biomin F 76.30 ± 11.44 mg/L; P^−^ Biomin C 2.623 ± 0.393 mg/L; Si^−^ 45S5-45.15 ± 6.77mg/L, F^−^ Biomin F^−^ 3.256 ± 0.651mg/L; Cl^−^ Biomin C 135.5 ± 20.3 mg/L after 45 min). Conclusions: These findings highlight the gels’ capacity to kickstart the remineralization process by delivering critical ions needed for enamel layer reconstruction. Further exploration in more dynamic, real-world conditions is recommended to fully ascertain their practical utility.

## 1. Introduction

Enamel demineralization in the oral cavity can occur due to long-term exposure to low pH. Such demineralization may be attributed to diets high in carbohydrates, whose breakdown products are acidic, and to teeth whitening treatments [1,2]. In their review, Pepla et al. [1] highlight the role of nano-hydroxyapatite in remineralization processes, suggesting its potential in counteracting the effects of acidic products from carbohydrate-rich diets. Moreover, alterations to the tooth surface can lead to hypersensitivity through the exposure of dentinal tubules [3]. Sari et al. [3] discuss the development of a hydroxyapatite nanoparticle-based gel aimed at enamel remineralization, which could effectively address the hypersensitivity issues caused by exposed dentinal tubules.

To counteract this adverse event, particularly early stage demineralization, it is essential to locally supply ample calcium cations and phosphate anions to achieve supersaturation. This leads to their re-deposition on the tooth surface, initiating the remineralization process and reducing hypersensitivity [4,5]. The study by Jafari et al. [4] on the application of bioactive glasses in dentistry supports the notion that such materials can significantly contribute to the remineralization process, thereby reducing hypersensitivity by occluding the exposed dentinal tubules.

According to hydrodynamic theory, another method to alleviate hypersensitivity involves using potassium ions, which disrupt neural conduction and thereby relieve pain. However, this relief is not long-lasting, leading many researchers to concur that a more enduring effect might stem from occluding the dentinal tubules [2,6,7]. Dionysopoulos et al. [7] review contemporary therapeutic approaches to dentin hypersensitivity, emphasizing the long-term benefits of tubule occlusion, particularly through the use of bioactive materials like calcium phosphate and nano-hydroxyapatite. This occlusion can be achieved with various ions, such as strontium or calcium ions and phosphate anions [8]. These substances are found in materials like amorphous calcium phosphate (ACP), tricalcium phosphate (TCP), nano-hydroxyapatite (nHA), bioactive glasses, among others. Recently, the most significant expectations have centered on two materials: hydroxyapatite (HA) and bioactive glasses.

Nano-hydroxyapatite can be synthesized using two primary methods: through the precipitation reaction of appropriate calcium salts and phosphates, and through the sol–gel reaction, which involves the hydrolysis of suitable organic compounds containing these elements [1,3].

Bioactive glasses can be produced using two distinct methods: by melting the appropriate oxides followed by rapid cooling and quenching (melt–quench method), or, similarly to n-HA, by hydrolyzing the suitable organic compounds. Depending on the method, products with different particle sizes and shapes can be obtained, which indirectly affects their biological properties. The unique remineralizing properties of bioactive glass have been acknowledged in dental research [9]. Notably, a study by Madan et al. highlighted how dentifrices with calcium-sodium-phosphosilicate, a key component of bioactive glass, enhance tooth remineralization by supplying essential minerals like calcium and phosphorus directly to the tooth surface, thereby aiding in the natural remineralization process that might be insufficient in maintaining strong enamel [9].

In the case of these materials, their size and potential criticality are very important, as they may affect the ability to deliver ions and the depth of ion penetration into the lesion. For this reason, different products have different mineralization effects [8,9]. Research has demonstrated the capacity of bioactive glass not only to deliver critical ions for remineralization but also to initiate the formation of hydroxy carbonate apatite (HCA), which closely resembles the mineral composition of natural teeth. This effect is pivotal for the in vitro remineralization of dental enamel, as detailed in studies exploring the interaction of bioactive glasses with human dental tissues [8,10,11].

Another method to reduce enamel demineralization involves fluorine-containing compounds replacing OH groups in hydroxyapatite crystals, creating less soluble fluorohydroxyapatite crystals. In this field, the most popular products are various types of toothpastes, mouthwashes, gels, varnishes, and others. Fluorine anions can be delivered from NaF, amine fluorides, and bioactive glasses. The integration of bioactive glass in dental care products, especially those containing fluoride, marks a significant advancement in efforts to enhance enamel demineralization resistance [10,11]. The dual function of bioactive glasses in releasing essential ions for remineralization while also providing fluoride ions contributes to the formation of fluorohydroxyapatite, a more resilient variant of the mineral, thus offering an added layer of protection against caries and erosion [12].

Additionally, fluorine ions can capture calcium cations from saliva, facilitating their adhesion into the HA structure. Fluoride itself has bacteriostatic properties, which inhibits the demineralization process. Fluorine ions show the greatest activity towards bacteria at acidic pH; at a very low concentration of 0.1 mM, it can completely stop glycolysis in intact Streptococcus mutans cells. The anti-caries effect of fluoride is complex and includes the effect on both bacteria and the formation of fluorohydroxyapatite in the mineral phases of mineral enamel. This compound has much lower solubility than the hydroxyapatite and antimicrobial actions of fluoride for oral bacteria [13,14].

The most popular method to deliver materials that can strengthen the tooth surface is through toothpastes, used for daily oral prophylaxis. However, the interaction between toothpaste and the tooth surface is short-lived, lasting only a from few seconds to 2–3 min [15].

During the whitening treatment, bleaching trays are frequently used, individually made for each patient. In the intervals between the application of subsequent doses of teeth whitening gels, patients are often instructed to use the gels to reduce tooth hypersensitivity connected with the bleaching process. Then, the contact of such substances with the tooth surface is prolonged and may last several hours. This duration is optimal for materials containing bioactive glass (BAG) to undergo the hydrolysis process, as well as the released calcium and phosphate ions, which can be incorporated into the open dentinal tubules or to harden the enamel surface, with the potential to be weakened by the tooth whitening treatment [16,17].

Therefore, it becomes advisable to create and test gels containing bioactive glasses, which could be used both in the dentist’s office and by the patient at home, both on a whitening tray and after cleaning teeth with toothpaste. A 20% concentration of bioactive glass was chosen for the gel preparation based on the findings in the article by Sari et al. [3], which used the same concentration of nano-hydroxyapatite.

There is information in the literature that bio-glass can release appropriate ions, but the quantity released over time can vary significantly depending on the composition of the matrix (gel) in which they are distributed [18]. Indeed, Madan et al. (2011) emphasized the unique remineralizing properties of bioactive glass, particularly its ability to provide essential minerals such as calcium and phosphorus to the tooth surface. This process enhances tooth remineralization beyond the natural capacity, crucial for maintaining strong enamel in the face of demineralization challenges [9]. This is particularly evident in the case of nano-HA-based gels, where both silica-based glycerol hydrogels [19] and carbomer-based gels [3] containing n-HA have been shown to repair damaged enamel [20]. However, it is challenging to find information on gels containing bioactive glasses in their composition and the rate at which specific ions are released from these materials. The need for comprehensive data on the ion release kinetics from bioactive-glass-containing materials is echoed by the findings of Ji-Hyun Jang et al. (2018), who investigated the effects of bioactive glass within resin composites on dentin remineralization. Their research could offer insights into the behavior of bioactive glasses in gel matrices as well, highlighting the importance of understanding the release patterns to optimize remineralization efficacy [21].

Consequently, the objective of these tests was to prepare suitable gels containing BAG and to evaluate their physical properties, including disintegration time, pH, ion release, consistency, viscosity, and the stability of these parameters over time.

The hypothesis presented at the beginning of this article posits that gels containing hydroxyethyl cellulose as a thixotropic agent and four different types of bioactive glasses can consistently release ions over time and remain stable during storage.

## 2. Materials and Methods

### 2.1. Materials Preparation

Bioactive glass samples were obtained from Cera Dynamic (Kent, England). The basic composition of the products is presented in Table 1. A sample of this material, which was also used in our previous work, is known to have an average particle size of d50 = 5 microns. The materials were utilized for testing as received, without any further processing [22]. All chemical reagents: glycerin 99%, sorbitol 70%, and hydroxyethyl benzoate, were purchased from Sigma Aldrich (Prague, Czech Republic).

For the tests, 5 gels with different BAGs were prepared, 300 g each (4 containing bioactive glass and one blank). A popular agent for creating oral gels is hydroxyethyl cellulose [20]. To prepare a single gel, 140 g of distilled water was used, to which 6 g of hydroxyethyl cellulose (Narosol 250, Aqualon™, Ashland, Delaware, WI, USA) was added. The material was stirred in a glass beaker using a magnetic mixer (300 rpm, Vevor SH-2, Vevor, Richmond, CA, USA) until the polymer was dissolved. A total of 60 g of glycerin and 30 g of sorbitol were added to the gel prepared in this way. Next, 0.3 g of the preservative hydroxyethyl benzoate (Sigma Aldrich, Prague, Czech Republic) was added to the mixture. The selection of these specific chemical reagents, especially hydroxyethyl benzoate as a preservative, plays a crucial role in maintaining the stability of the gel matrix. This stability is essential for ensuring the controlled release of ions from the bioactive glass over time, contributing to the material’s remineralization capabilities [22]. After 15 min of stirring, 60 g of the appropriate bioactive glass was added to the solution. Finally, 3 g of Aerosil A 200 (Evonik, Essen, Germany) and mint flavor were added (Stringer Flavor, Tring, UK). White-colored gels were obtained and placed in a vacuum dryer (GOLDBRUNN 1450, Goldbrunn Therm, Obersdorf, Germany) at a pressure of 0.3 bar to remove any trapped air from the mixture for 60 min. Next, the materials were filled into 60 mL PE tubes and closed. The blank gel (placebo), devoid of bioactive glass content, was prepared using the same method and used as a control for further tests [23]. Additionally, for stability testing, samples were placed in a laboratory dryer for a period of 2 months at 37 ± 2 °C. Stability testing under these conditions is crucial to simulate the shelf-life of the gels and predict their performance over time. 

### 2.2. Disintegration Time

One essential requirement for a gel is its disintegration time, which indicates the duration the material can adhere to the tooth surface. This test is commonly employed for various medicinal products in gel form [24]. The methodology for this investigation was adapted from the article and the American Pharmacopeia [25,26]. A magnetic stirrer was placed in a 100 mL glass beaker, which was then filled with 40 mL of distilled water. A total of 0.1 g of the test gel was applied to a microscope slide’s surface, and the entire assembly was submerged in water (Figure 1). This quantity of the test material was chosen to represent the amount that might come into contact with a single tooth’s surface.

The speed of the magnetic mixer (IKA Breisgau, Germany) was set to 100 rpm, and the time was measured using a stopwatch. The test concluded when no gel remnants remained on the surface of the slide. Five samples from each composition were tested, and averages were calculated, resulting in a total of 75 measurements across 3 time intervals (0–2 months).

### 2.3. Measuring the pH of the Solution

For the material to effectively initiate the remineralization process, an appropriate pH is required, which should increase the pH above 5.5. For this purpose, the solutions obtained in the previous test were subjected to a pH measurement shortly after the disintegration of the gel. 

Prior to the study, the pH meter was calibrated with a ready-made buffer solution at pH = 7 (Sigma Aldrich, Praha, Czech Republic). Tests were performed with a Mettler Toledo pH7110 SET instrument (Mettler Toledo sro, Prague, Czech Republic) equipped with a SenTix^®^ 81 electrode and 3 mol/L KCl buffer solutions. The instrument required daily calibration, which was performed prior to each test [23,27,28]. Five samples from each composition were tested, and averages were calculated, resulting in a total of 75 measurements across 3 time intervals (0–2 months). The distilled water used for testing had a pH of 5.43.

### 2.4. Material Consistency

This test is commonly employed for pasty materials, allowing for easy determination of how the material will behave after being squeezed out of the tube and whether it will remain on the applied surface [23].

A sample of the material with a volume of 0.5 cm³ was placed on the surface of a 40 mm × 40 mm glass slide covered with PE foil. The testing gel was covered with a second layer of PE foil and a thin glass (mass 70 g). The assembly was then loaded with a 500 g weight for 5 s, after which the diameter of the resultant circle was measured. The larger the diameter, the more fluid the material is.

Five samples from each composition were tested, and averages were calculated, resulting in a total of 75 measurements across 3 time intervals (0–2 months).

### 2.5. Viscosity of Materials

Samples of gel materials were measured using an Anton Paar plate rheometer (Nagano Park, Prague, Czech Republic), as shown in Figure 2. A 25 mm diameter spindle and a viscosity testing program, both provided by the rheometer manufacturer, were used for the tests. The measurements were conducted at two temperatures: 23 °C and 37 °C. These temperatures were selected to evaluate the material properties during storage (at room temperature, 23 °C) and post-application in the oral cavity. The choice of these specific temperatures reflects the practical conditions under which the gels are used and stored, aligning with findings by Makanjuola and Deb (2023) [18]. 

Five samples from each composition were tested at both temperatures, and averages were calculated, resulting in a total of 150 measurements across 3 time intervals (0–2 months) (Figure 2).

### 2.6. Determination of Elements Release

The kinetics of release of elements (Ca, P, Si, Na, F^−^, Cl^−^) from the 4 materials obtained (Biomin C, Biomin F, 45 S5, S53 P4) was carried out in ultrapure water at pH 5.5 (Sigma Aldrich, Poznan, Poland). A total of 1 g of each dental gel was transferred to microscope slides and then placed in polypropylene dishes with 20 mL of ultrapure water (Figure 3). The prepared samples were placed on an orbital shaker (60 rpm). The number of replicates was prepared to stop the process after 5, 10, 15, 20, 25, 30, 35, 40, and 45 min, separately. A polypropylene dish containing a watch glass and water was used as a control (135 samples totally). After the specified time, the samples were filtered and submitted for analysis [26,27].

#### 2.6.1. Mineralization of Dental Gels 

Decomposition of dental gel (4 samples) was carried out by two-stage microwave-assisted mineralization, in a closed, wet system using the START D microwave decomposition system (Milestone, Sorisole, Italy). Samples weighing approximately 0.1 g each were placed in Teflon dishes. In the first step, 2 mL each of royal water (0.5 mL of nitric acid and 1.5 mL of hydrochloric acid) and 1 mL of hydrofluoric acid (all Suprapur purity acids, Merck, Darmstadt, Germany) were added to the samples. Mineralization was carried out for 10 min, at a temperature of 100 °C and an oven power of 1000 W. In the second stage, 10 mL of boric acid (Merck, Darmstadt, Germany) was added to Teflon dishes with premineralized materials. The process was carried out for another 35 min, at a temperature in the range of 100–200 °C and an oven power of 1000 W. After the process, the cooled mineralizates were transferred to bottles made of HDPE material and diluted to a weight of about 50 g.

#### 2.6.2. Elemental Composition Analysis

The elemental composition (Ca, P, Si, Na) of extracts and dental materials was analyzed by inductively coupled plasma atomic emission spectrometry (ICP-OES), using an iCAP 6500 Duo horizontal and vertical plasma optical spectrometer (Thersmo Fisher Scientific, Waltham, MA, USA). Extracts and mineralized dental gels were analyzed using validated test methods with compensation for the influence of matrix effects at the Chemical Laboratory for Multi-elemental Analysis, accredited by the Polish Accreditation Center (AB 969).

#### 2.6.3. Analysis of Fluoride and Chloride Concentrations

The fluoride and chloride concentrations of the extracts were analyzed by ion chromatography, using a Dionex ICS 1100 ion chromatograph (Thermo Fisher Scientific, Waltham, MA, USA). Dental material extracts were directly injected through a sterile 0.2 µm syringe filter onto the chromatography column. For dental gels, extraction processes were carried out prior to measurement according to the methodology given below:Chloride extraction—100 mL of ultrapure water was added to 1 g of dental gel, and then shaken for 30 min. After this time, the samples were filtered into a volumetric flask (100 mL).Fluoride extraction—50 mL of HCl (concentrated HCl:H_2_O 1:1) was added to 3 g of dental gel, and then the mixture was heated at 100 °C for 1 h. After cooling, it was filtered into a volumetric flask (250 mL).

The prepared extracts were injected through a sterile filter onto the column.

### 2.7. Adhesion of Gels

To assess adhesion, in vitro tests were conducted between two glass plates. Following the method outlined by Budi et al., 0.5 g of gel was placed between glass plates weighing 50 g and measuring 50 × 50 mm; a 1000 g weight was then applied for a period of 5 min [28]. Subsequently, the plates were mounted in a Shimadzu stretch testing instrument (Kyoto, Japan), and the resulting bond was stretched at a speed of 5 mm/min. The test concluded when the two surfaces bonded by the gel peeled apart from each other. Five repetitions, totaling 75 tests, were performed for each gel type.

### 2.8. Gels under a Microscope

Gel samples were placed on microscope slides and analyzed under 400× magnification using a Kern OB-1 microscope (Kern and Son, Baden-Württemberg, Germany). The visualization of gels under high magnification is crucial for assessing the microstructure and potential for bioactive glass particles to contribute to remineralization by acting on enamel demineralization inhibition [29,30,31].

### 2.9. Stability of Materials over Time

Aging tests were conducted by storing the gels in tightly sealed tubes within a laboratory dryer at 37 ± 2 °C for 2 months to determine the material’s behavior over time. Every 30 days, the samples were removed from the dryer and, once the material reached room temperature, the disintegration time, pH, consistency, and viscosity were assessed. 

### 2.10. Statistical Analysis

The results were statistically analyzed using a repeated measures ANOVA at a significance level of 0.05 to compare the stability of each gel over time. Additionally, Multiple Correction: Bonferroni was used, utilizing a free calculator provided by Statistics Kingdom (Retford, UK) to assess the stability of each gel over time. 

## 3. Results

The results obtained from the gels containing bioactive glass (BAG) are presented in the subsequent tables: Table 2 for 45S5, Table 3 for S53P4, Table 4 for Biomin C, Table 5 for Biomin F, and Table 6 for the blank gel.

The disintegration time of the gel with the glass 45S5 increased when stored at 37 °C, while the pH gradually decreased, with minor changes in consistency. Simultaneously, the viscosity at 23 °C increased. 

The disintegration time increased for the samples with S53P4 glass with storage at 37 °C, whereas adhesion decreased, accompanied by a slight reduction in consistency. The viscosity measured at 23 °C increased during the accelerated aging tests. 

When we stored the samples of the gel with Biomin Fat 37 °C, the disintegration time increased, along with a slight rise in gel consistency. Conversely, the viscosity at 23 °C decreased during storage at elevated temperatures.

The gels with Biomin F glass had shorter disintegration time s when stored at 37 °C, and the pH increased along with consistency after 1 month. Initially, the viscosity decreased, and then it returned to its original value after a month. 

The control gel’s disintegration time increased when stored at 37 °C, with negligible change in pH. Viscosity increased significantly, impacting adhesion. Consistency decreased, while viscosity experienced an approximate 400 (mPa/s) increase after 1 month, and then it stabilized. 

An exemplary change was in the viscosity of a 45S5 gel measured using an Anton Paar rheometer, which was stored at a temperature of 37 °C for 2 months (Figure 4). A comparison of the viscosity of all the performed gels is shown in Figure 5.

The appearance of individual gels in the microscopic image is presented in Figure 6a–d.

By examining the gels under a microscopic image at a magnification of 400×, it was possible to observe larger glass particles with a size of 10 μm. Additionally, depending on the type of glass, they may have had more rounded shapes: S45P4 Biomin C and shapes with more sharp edges: 45S5 and Biomin F.

After 22 h, a white precipitate, which was the remains of hydrolyzed glass, was visible at the bottom of the vessel where the gel was dissolved (Figure 7). 

During hydrolysis in aqueous water, the bioactive glasses contained in the gels released specific elements. The results of this are presented in Table 7.

The higher concentration of elements released into the solution by partially hydrolyzed gels containing bioactive glass increased when the material was dissolved in water after 45 min. The sample containing Biomin C released the calcium (104.8 ± 15.7 mg/L), which was consistent with the basic chemical composition of this glass.In the case of phosphorus, 2.623 ± 0.393 mg/L was determined for Biomin C glass; the percentage of this element in the glass was approximately 2.4%. The situation with the release of chlorine anions in this glass was similar, 135.5 ± 20.3 mg/L at a content of over 10%.The highest content of sodium (89.41 ± 13.41 mg/L) and silicon (45.15 ± 6.77 mg/L) in the sample was determined for 45S5 glass. Both of these elements constituted 17% and 14% of this glass, respectively.Biomin F can release the largest amount of fluorine anions, 3.256 ± 0.651 mg/L, having 2% of this element in its composition.

## 4. Discussion

The hypothesis introduced at the beginning of this paper was validated. Gels formulated with hydroxyethyl cellulose as a gelling agent and bioactive glasses were demonstrated to release ions. Furthermore, these gels exhibited stability over a two-month period in accelerated aging tests conducted at 37 ± 2 °C. The effectiveness of ultrafine bioactive glass particles in remineralizing human dentin, showcasing significant mineral deposition within the treated areas, underscores the potential of these gels in dental applications [32].

Hydroxyethyl cellulose (HEC) is non-toxic and possesses excellent water-binding capabilities. Utilizing 2% HEC in formulations achieves the desired viscosity, and the gelling properties can be enhanced by incorporating a co-solvent or a solubility enhancer. Glycerin, often incorporated as a co-solvent in gel formulations at concentrations up to 80%, is favored for its hydrophilic and hydrophobic groups. This allows it to dissolve hydrophobic compounds efficiently, exhibit good water solubility, and maintain low toxicity. Consequently, many oral gels containing therapeutic or active substances comprise these two critical components [33].

Upon application in the oral cavity, the dental gel interacts with water, leading to swelling, solubilization, and the gradual release of active substances. For these experiments, the active substances are bioactive glasses. Greenspan and Hench (2013) highlighted the effectiveness of bioactive-glass-containing gels in tooth remineralization and pain desensitization, emphasizing their stability and reactive nature under physiological conditions [33].

Maçon et al. observed a rise in pH in water solutions upon the addition of a paste with bioactive glass. After approximately 8 h, the pH reached 8, and after 24 h, it escalated to 11. The variance in pH elevation could be attributed to the BAG particle size and the rate of hydrolysis. However, the specific composition of the glass from GlaxoSmithKline was not disclosed by the authors [34]. This observation aligns with the findings of Fernando et al. (2017), who systematically reviewed the bioactive glass’s role in dentin remineralization, underscoring its ability to elevate pH levels and foster a conducive environment for mineral deposition [35].

This article suggests another factor that may impede ion release from toothpaste: the organic matrix encapsulating BAG particles, which hampers the dissolution/precipitation mechanism. This could lead to changes in ion mobility due to the presence of numerous organic compounds. The rapid change in pH as detailed by Siekkinen et al. (2023) is crucial for the initiation of the remineralization process, setting the stage for the restoration of dental enamel through the deposition of essential minerals [36].

Akbarzade et al. report that the dissolution of BAG in water elevates the pH due to an increase in calcium and sodium ion concentrations. It is posited that toothpastes containing bioactive glass, which forms amorphous calcium phosphate upon hydrolysis, aid in the remineralization of tooth surfaces [37]. However, this requires glasses with an appropriate particle size. The in vitro study on deciduous teeth by Zhang et al. (2021), where 45S5 bioactive glass showed pronounced mineral gain and surface density in remineralizing early carious lesions, exemplifies this requirement [38].

In the current tests, a similar pH value of 8 was reached just 30 min after the gels had completely dissolved in distilled water. This might be attributed to the gel’s distinct chemical composition, primarily comprising glycerin and sorbitol. Water facilitates the initial hydrolysis of the glass within the gel prior to its application. The pH results we obtained are promising, as an increase in pH to no more than 9.70 (for the gel with Biomin F) is acceptable and does not irritate soft tissues [39,40]. The study by Dong et al. (2011) focused on the in vitro remineralization of human dental enamel by bioactive glasses, highlighting the significant role of bioactive glass’s chemical composition in promoting enamel remineralization through the formation of a hydroxycarbonate apatite (HCA) layer similar to natural enamel [41].

The largest pH changes over time were observed for the 45S5 gels (from 9.34 to 8.9) and for the S53P4 samples (from 9.04 to 9.5), which are not significant changes. Bioactive glass in an aqueous environment immediately begins to react in three different ways: leaching and exchange of cations, hydrolysis of the SiO_2_ matrix, and precipitation of calcium and phosphates, leading to the formation of an appetite layer on the surface of the silica scaffold, which is one of the by-products from the reaction of BAG with water. Fernando et al. (2017) systematically reviewed the efficacy of bioactive glass in dentin remineralization, confirming its multifaceted action in an aqueous environment, which includes ion exchange, matrix hydrolysis, and the formation of a protective apatite layer, thus providing a substantial basis for its use in dental remineralization applications [35].

The initial exchange of ions, Na^+^ from the surface and H^+^/H_3_O^+^ from water, as well as the de-alkalinization of the glass surface layer, proceeded quite quickly, within a few minutes of the glass’s contact with an aqueous medium. The surface changes and sodium loss caused the local decomposition of the silica network, resulting in the formation of silanol groups (Si(OH)_4_), which facilitated the formation of three-dimensional structures. The base-catalyzed hydrolysis of Si–O–Si bonds within the glass structure occurred 3–6 h after the glass was introduced into the body. At the end of this process, a layer of calcium phosphate can bond to the enamel [42].

In commercial gels containing various active substances, it is recommended to leave the material on the tooth surface for a minimum of 3–5 min and avoid eating for 30 min [43]. Therefore, the gel containing bioactive glass can disintegrate in the oral cavity within 20–30 min. The disintegration time for the gels obtained in these studies ranged from 38 to 51 min. This time varied among the individual gels, even with the same HEC content, indicating that interactions may occur between the cellulose and the ions released from the bioactive glasses. The variation in disintegration time observed in our studies aligns with findings from Vollenweider et al. (2007), where ultrafine bioactive glass particles were shown to influence the disintegration time of gels through their interaction with the gel matrix, thus affecting the overall bioactivity and potential for dental tissue remineralization [32].

The prolonged dissolution time may result from the slow relaxation of HEC macromolecules in the solution during storage or the slow hydrolysis of bioactive glasses, leading to the formation of a SiO_2_ gel that further slows the glass’s hydrolysis. Upon complete hydrolysis of the gel, a white silica precipitate is visible at the bottom, as depicted in Figure 7 under a microscope [33,34]. When storing the materials at an elevated temperature for 2 months, an increase in disintegration time from 40 to 50 min was observed for gels containing Biomin C, 45S5, and S53P4. However, a gel containing Biomin F dissolved faster in water, from 45 to 32 min, which is still clinically acceptable.

Bioactive glass in an aqueous environment immediately begins to react in three distinct ways: leaching and exchange of cations, hydrolysis of the SiO_2_ matrix, and precipitation of calcium and phosphates, leading to the formation of an appetite layer on the surface of the silica scaffold [44,45,46]. This rapid interaction with the environment is crucial for dental applications, as demonstrated by Dong et al. (2011), who explored the in vitro remineralization of human dental enamel by bioactive glasses, finding significant remineralization potential and suggesting the effectiveness of bioactive glass in dental repair and maintenance [41].

The higher concentration of the gelling substance led to the longer the dissolution time in water. This concept was further examined by Vollenweider et al. (2007), who investigated the remineralization potential of ultrafine bioactive glass particles on human dentin. Their findings support the notion that the physical properties of the gel, such as viscosity, play a significant role in its therapeutic effectiveness, with optimal ranges promoting better adhesion and interaction with dental tissues [32].

The viscosity of most gels (45S5, S53P4, Biomin F, and Blank) increases over time when stored at a temperature of 37 °C, which can be attributed to two main factors: the slow migration of water vapor through the packaging and phenomena occurring within the HEC itself. Further research by Greenspan and Hench (2013) on bioactive glass for tooth remineralization emphasizes the critical role of storage conditions, including temperature, in preserving the bioactive properties of gels intended for dental applications, thus impacting their efficacy in remineralization processes [33].

The properties of hydroxyethyl cellulose solutions are also influenced by pH changes. In an acidic environment, this polysaccharide undergoes hydrolysis, while in an alkaline environment, its hydrolysis is preceded by oxidation. This process ultimately leads to the degradation of HEC and an increase in the content of highly polar carboxyl groups over time (Blažková et al., 1999) [47]. The pH-dependent behavior of hydroxyethyl cellulose and its impact on gel properties was explored in depth by Vollenweider et al. (2007), who noted that pH adjustments could significantly alter the sol–gel transition of bioactive-glass-containing gels, potentially enhancing their remineralization capacity [32].

In aqueous solutions, cations and anions from electrolytes, like those in hydrolyzed bioactive glasses, can bind to oxygen and hydrogen atoms in solvent molecules and to the free -OH groups of the glucose unit of the polysaccharide, forming ionic complexes. Zhang et al. (2021) provided a quantitative analysis in their study, showing that the inclusion of 45S5 bioactive glass at concentrations up to 6% could lead to substantial changes in gel viscosity, which in turn influences the gel’s ability to facilitate dental remineralization through enhanced ion release and surface interaction [38].

Viscosity tests have demonstrated that samples undergoing a heating process at elevated temperatures exhibit an increase in viscosity. An additional increase in viscosity over time may be attributed to the influence of cations from the first group of the periodic table, with divalent cations exerting a lesser impact on the increase in viscosity over time, as confirmed by Dong et al. (2011), who quantitatively assessed the impact of various cations on the viscosity and overall performance of bioactive-glass-containing gels [41].

Waly et al. emphasize the importance of the appropriate concentration of the cross-linking agent, hydroxyethyl cellulose, as it must simultaneously enhance the cohesion of the hydrogel while keeping its viscosity low enough to allow for the gel’s extrusion from the tube. Storing the gels for a period of 2 months in a laboratory dryer at a temperature of 37 °C leads to changes in their viscosity; however, these changes are not substantial, and the gel still retains enough viscosity to adhere effectively to the tooth surface [43].

An essential parameter is the adhesive strength of a given gel to a surface [30,31,39]. In the preliminary tests, we used a basic model that involved placing the gel between two glass plates and measuring the force required to separate them.

This approach significantly simplifies the in vivo behavior of the substance; thus, future research is necessary. The energy needed to separate two plates with a gel containing bioactive glass (Biomin F, Biomin C, and S53P4) is higher than that for the gel containing HEC alone. The results of the gel’s adhesion to the glass plate suggest that it may depend on two factors: the viscosity of the gel (with higher viscosity resulting in greater adhesion) and the type of glass within the gel. Zhang et al. (2021) suggest that the specific composition and concentration of bioactive glass within gels could significantly influence their adhesive properties and effectiveness [38].

Furthermore, after storing the sample at an elevated temperature, this value doubles compared to the initial value for all samples except S53P4. Greenspan and Hench (2013) discussed the physical properties of gels, such as viscosity and adhesive strength, may be further enhanced by the bioactive glass’s interaction with dental tissues, even after exposure to elevated temperature [33].

Maçon et al. (2015) measured the ion release from toothpaste containing 5% BAG [34] (aNovaMin^TM^ (46.1 mol% SiO_2_, 26.9 mol% CaO, 24.4 mol% Na_2_O, 2.6 mol% P_2_O_5_)). While dissolving the paste, the authors observed the release of calcium and phosphorus, which decreased over time, and silicon, whose concentration increased within 3 days.

Waly (2018) [48] investigated the release of specific ions from a gel composed of a chitosan/hydroxyethyl cellulose matrix and amorphous calcium phosphate/casein phosphopeptide. These authors also observed that HEC-based gel can release calcium and phosphate ions over time.

Hmood et al. (2018) tested the ion release from various glasses [49]. Here, the amount of isolated Ca, Na, and P elements increased over time up to 100 h, when the experiment was carried out using a buffer with pH = 7.4.

In the case of our tests, we obtained high results of an increase in the number of isolated elements within 45 min. This may indicate the initial hydrolysis of bio-glass in gel solutions containing 30% water. The concentration of Ca (104.8 ± 15.7 mg/L), phosphorus (2.623 ± 0.393 mg/L) and chlorine anion (135.5 ± 20.3 mg/L) increased the fastest in samples containing Biomin C. S53P4 (89.41 ± 13.41 mg/L), 45S5, and Biomin F glasses released a large amount of sodium in a short time period. The release of silicon in all samples occurred at the same rate, reaching a concentration of 35–45 mg/L after the end of the test.

However, the conducted research has certain limitations. The developed gels require further testing to assess their biological properties, including cytotoxicity in cell cultures, as well as dental tests to verify their capability to penetrate dentin tubules and thereby alleviate hypersensitivity [50,51]. Simila and Boccaccini (2022) reviewed sol–gel bioactive-glass-containing biomaterials for restorative dentistry, highlighting the importance of further in vitro and in vivo studies to ascertain the full range of biological properties of these materials, including their ability to penetrate dentin tubules and reduce hypersensitivity [52]. Moreover, subsequent phases will necessitate clinical trials to evaluate the in vivo effectiveness of these properties.

### Future Perspectives

The study on hydroxyethyl cellulose-based gels combined with bioactive glass (BAG) marks a significant step forward in dental health research, particularly in the areas of remineralization and hypersensitivity treatment. The gels’ capacity to release essential ions and raise pH levels highlights their potential in dental care. Future investigations are poised to explore various BAG compositions to enhance the gels’ remineralization effectiveness, with existing research suggesting that the specific makeup of bioactive glass plays a crucial role in its efficacy [41].

Further exploration into innovative bioactive materials, such as nanoparticles or advanced ceramics, could improve the remineralization properties of dental gels, promising groundbreaking advancements in dental solutions [52]. The transition from laboratory findings to real-world applications is essential, underscoring the importance of in vivo studies and clinical trials in assessing the gels’ effectiveness and safety in dental applications [35].

Tailoring gel formulations to address particular dental conditions could result in more precise and potent treatments, potentially integrating these gels with other dental therapies for comprehensive care. Exploring new methods for applying these gels, like dental trays or inclusion in dental hygiene products, might make treatments more practical and effective [38].

Long-term assessments of these gels’ impact on dental health are crucial, including potential side effects and interactions with other dental materials and treatments. Future research should also take into account the environmental and economic aspects of producing and using these bioactive gels to ensure they are accessible and environmentally sustainable. 

## 5. Conclusions

This study on gels made from hydroxyethyl cellulose and infused with bioactive glass (BAG) for dental remineralization offers encouraging outcomes. These gels, with a 20% composition of different BAG types such as 45S5, S53P4, Biomin F, and Biomin C, alongside a control variant, have proven effective in releasing vital ions such as calcium, phosphate, and silicon, leading to an increase in pH. This pH elevation suggests an environment conducive to the remineralization process.

Key observations from this research include effective ion release by the gels, leading to a pH increase that indicates an environment favorable for enamel repair. The gels’ dissolution in water within 30–50 min post-application suggests their suitability for therapeutic use, as they remain in contact with the dental surface long enough to deliver therapeutic effects. Additionally, the viscosity range of the gels, from 3269 ± 18 mPa/s for Biomin F to 6337 ± 24 mPa/s for 45S5, indicates adequate viscosity for effective adhesion to the tooth surface, enhancing treatment efficacy.

Hydroxyethyl-cellulose-based gels with bioactive glass show promise in treating dental hypersensitivity and aiding enamel remineralization. Their ion-releasing capability, pH elevation, and appropriate viscosity and adhesion characteristics highlight their potential as effective dental care products.

## Figures and Tables

**Figure 1 jfb-15-00119-f001:**
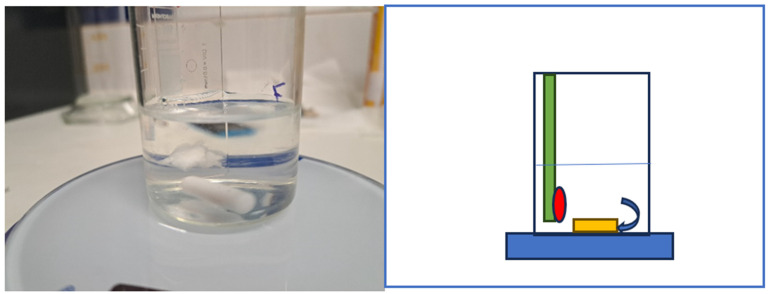
Method the determination of disintegration time; (blue) magnetic stirrer, (yellow) magnet of magnetic stirrer, (green) microscopic glass slab, (red) gel attached to glass slab.

**Figure 2 jfb-15-00119-f002:**
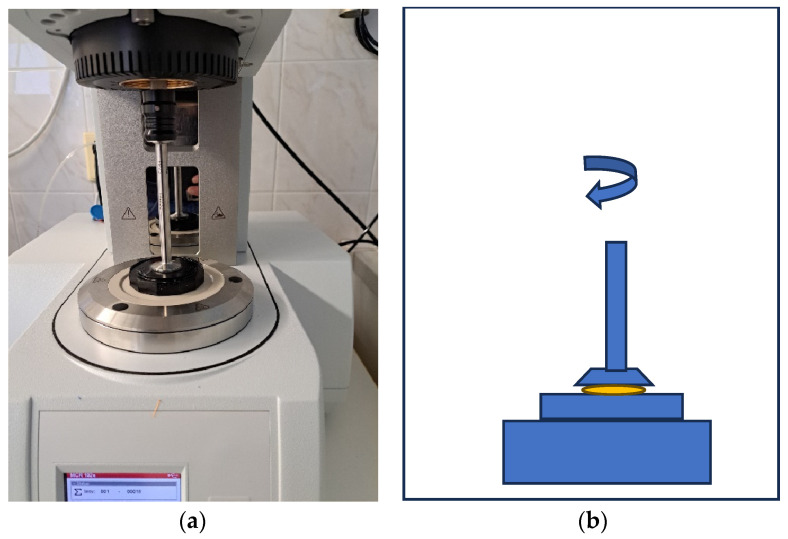
(**a**) The measuring geometry of the rheometer. (**b**) Measurement schema; (blue) elements of a plate rheometer, with thermostatic temperature; (yellow) gel sample, 1 mm thick.

**Figure 3 jfb-15-00119-f003:**
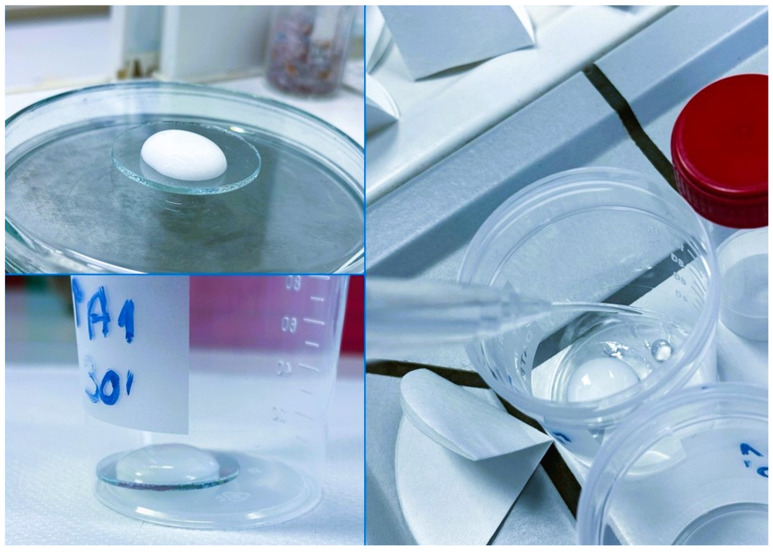
Preparation of the gel samples for the kinetics of elements release.

**Figure 4 jfb-15-00119-f004:**
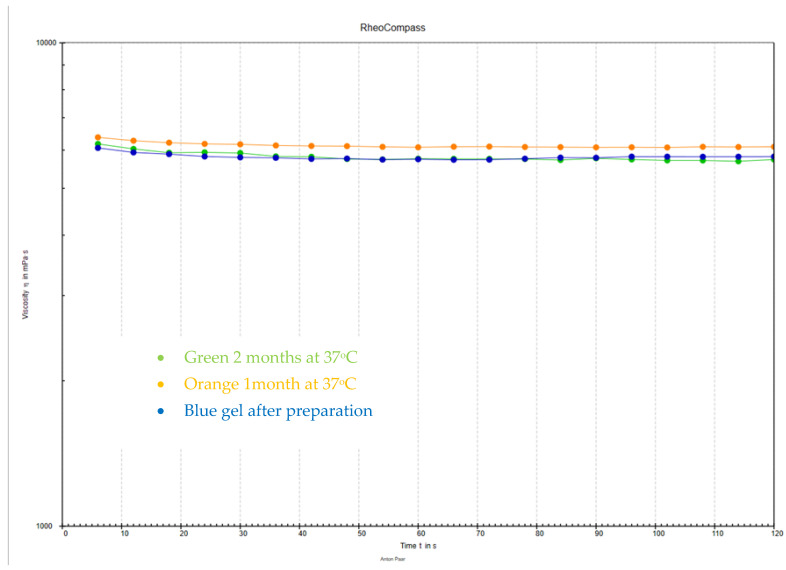
The change in viscosity for a 45S5 gel stored at 37 °C over 2 months, aligning with the trends discussed in the above studies.

**Figure 5 jfb-15-00119-f005:**
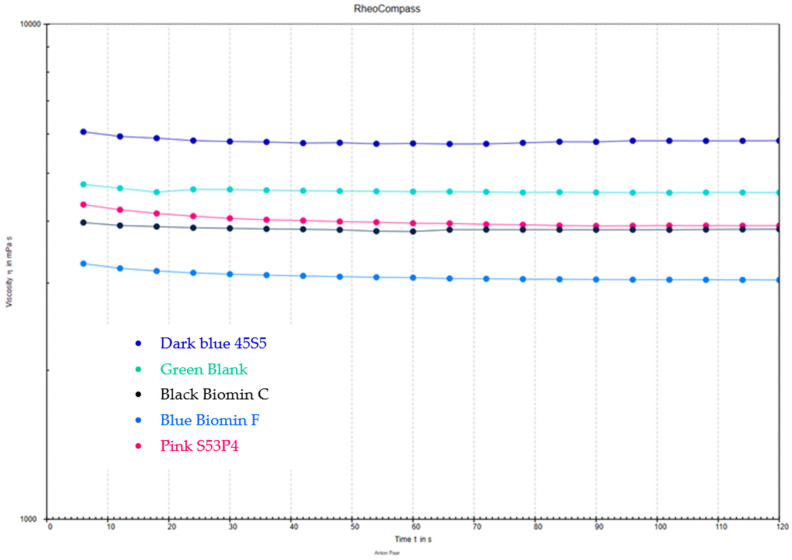
Viscosity curve showing the comparison between individual gels after their preparation.

**Figure 6 jfb-15-00119-f006:**
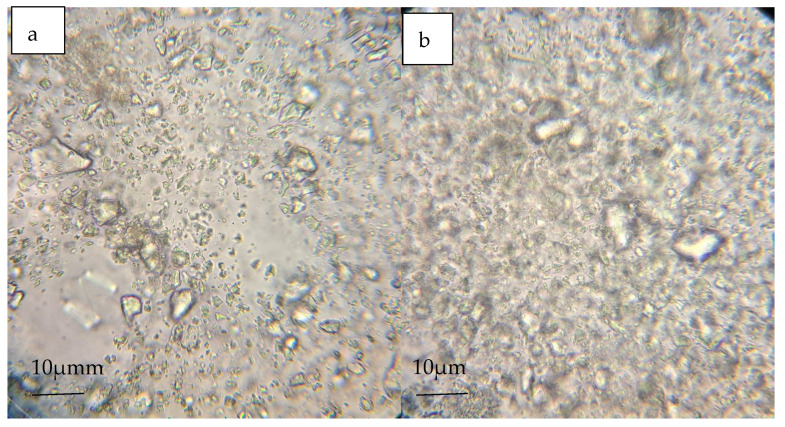
Gels after microscope examination: (**a**) S53P4; (**b**) 45S5; (**c**) Biomin F; (**d**) Biomin C.

**Figure 7 jfb-15-00119-f007:**
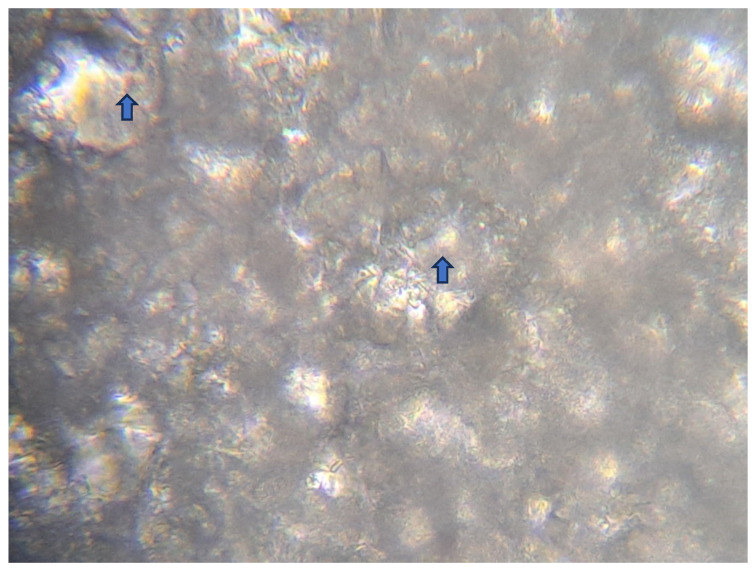
The precipitate collected after water hydrolysis of a gel containing 45S5 bioactive glass. Visible remains of crystalline glass particles (blue arrows)surrounded by an amorphous SiO_2_ gel.

**Table 1 jfb-15-00119-t001:** The composition of BAGs used in this study according to the information provided by the supplier. Values in % by weight.

	SiO_2_	P_2_O_5_	CaO	Na_2_O	CaF_2_	CaCl_2_
S53P4	53.8%	1.7%	21.8%	22.7%	0	0
Biomin F	36–40%	4–6%	28–30%	22–24%	1.5%–3.0%	0
45S5	4.1%	2.6%	26.9%	24.4%	0	0
Biomin C	30.3–31.8%	5.0–5.3%	44.1–46.3%	0	0	16.7–20.6%

**Table 2 jfb-15-00119-t002:** Stability of the gel with glass 45S5.

	After Preparation	1 Month Stability at 37 ± 2 °C	2 Months Stability at 37 ± 2 °C
**Disintegration Time (min)**	39 ± 5.0 **	40 ± 1.0 **	50.6 ± 2.2 **
**pH**	9.34 ± 0.03 **	9.00 ± 0.10	8.9 ± 0.10
**Consistency (mm)**	44 ± 2	45 ± 2	47 ± 1 *
**Viscosity (mPa/s) at 23 °C**	5804 ± 27 *	6140 ± 23 *	6337 ± 24 *
**Viscosity (mPa/s) at 37 °C**	5213 ± 21 *	4716 ± 22 *	4921 ± 49 *
**Adhesion (Pa)**	0.25 ± 0.04 *	0.38 ± 0.03 *	0.53 ± 0.04 *

* Statistically significant for *p* < 0.05 for the stability of the gel over time. (T = 0 → T = 2 months). ** Statistically significant for *p* < 0.01 for the stability of the gel over time. (T = 0 → T = 2 months).

**Table 3 jfb-15-00119-t003:** Time influence on the gel properties with S53P4 glass.

	After Preparation	1 Month Stability at 37 ± 2 °C	2 Months Stability at 37 ± 2 °C
**Disintegration Time (min)**	39 ± 0.5	40 ± 1.5	51.8 ± 2.2 *
**pH**	9.04 ± 0.10	9.10 ± 0.10	9.5 ± 0.00 *
**Consistency (mm)**	52 ± 1	51 ± 2	48 ± 2 **
**Viscosity (mPa/s) at 23 °C**	3995 ± 24 *	4522 ± 15 *	4864 ± 23 *
**Viscosity (mPa/s) at 37 °C**	3505 ± 11 *	3687 ± 8 *	4033 ± 16 *
**Adhesion (Pa)**	0.65 ± 0.04 *	0.54 ± 0.03	0.54 ± 0.03

* Statistically significant for *p* < 0.01 for the stability of the gel over time. (T = 0 → T = 2 months). ** Statistically significant for *p* < 0.05 for the stability of the gel over time. (T = 0 → T = 2 months).

**Table 4 jfb-15-00119-t004:** Stability of the samples with Biomin C.

	After Preparation	1 Month Stability at 37 ± 2 °C	2 Months Stability at 37 ± 2 °C
**Disintegration Time (min)**	38 ± 1	38 ± 1	50.6 ± 2.2 *
**pH**	9.51 ± 0.08	9.60 ± 0.10	9.5 ± 0.10
**Consistency (mm)**	50 ± 1	50 ± 1	54 ± 1 *
**Viscosity (mPa/s) at 23 °C**	3860 ± 10 *	3667 ± 18	3673 ± 14
**Viscosity (mPa/s) at 37 °C**	3191 ± 12	3137 ± 13	3381 ± 12 *
**Adhesion (Pa)**	0.32 ± 0.04 **	0.38 ± 0.02 **	0.43 ± 0.04 **

* Statistically significant for *p* < 0.01 for the stability of the gel over time. (T = 0 → T = 2 months). ** Statistically significant for *p* < 0.05 for the stability of the gel over time. (T = 0 → T = 2 months).

**Table 5 jfb-15-00119-t005:** The effect of time on a sample of gel containing Biomin F bioactive glass.

	After Preparation	1 Month Stability at 37 ± 2 °C	2 Months Stability at 37 ± 2 °C
**Disintegration Time (min)**	45 ± 3	39 ± 3	32.2 ± 1.8
**pH**	9.50 ± 0.01	9.40 ± 0.10	9.7 ± 0.00
**Consistency (mm)**	51 ± 2 *	55 ± 1	55 ± 1
**Viscosity (mPa/s) at 23 °C**	3043 ± 32 *	2968 ± 23 *	3269 ± 18 *
**Viscosity (mPa/s) at 37 °C**	2659 ± 14	2473 ± 16 *	2652 ± 14
**Adhesion (Pa)**	0.67 ± 0.03	0.68 ± 0.02	0.73 ± 0.06

* Statistically significant for *p* < 0.01 for the stability of the gel over time. (T = 0 → T = 2 months).

**Table 6 jfb-15-00119-t006:** Control gel without bioactive glass.

	After Preparation	1 Month Stability at 37 ± 2 °C	2 Months Stability at 37 ± 2 °C
**Disintegration Time (min)**	60.20 ± 3.49	60.60 ± 2.98	67.20 ± 2.80 **
**pH**	6.10 ± 0.04	6.11 ± 0.04	6.01 ± 0.06
**Consistency (mm)**	55 ± 1	55 ± 2	52 ± 2 *
**Viscosity (mPa/s) at 23 °C**	4608 ± 19 *	5051 ± 16	5043 ± 16
**Viscosity (mPa/s) at 37 °C**	4153 ± 20 *	4601 ± 14 *	4316 ± 13 *
**Adhesion (Pa)**	0.27 ± 0.04 *	0.38 ± 0.03 *	0.51 ± 0.06 *

* Statistically significant for *p* < 0.01 for the stability of the gel over time. (T = 0 → T = 2 months).** Statistically significant for *p* < 0.05 for the stability of the gel over time. (T = 0 → T = 2 months).

**Table 7 jfb-15-00119-t007:** Concentration of Na, Ca, P, and Si elements and Cl^−^ and F^−^ ions released from gel samples as a result of contact time with water (5–45 min).

Samples	Time	Ca			Na			P			Si			F^−^			Cl^−^		
	min	mg/L			mg/L			mg/L			mg/L			mg/L			mg/L		
(1 g sample + 20 mg/L water)																			
BIOMIN C	5′	48.81	±	7.32	6.126	±	0.919	0.584	±	0.088	12.76	±	1.91	0.2656	±	0.0531	76.30	±	10.09
	10′	59.54	±	8.93	6.027	±	0.904	1.005	±	0.151	15.40	±	2.31	0.3706	±	0.0741	80.80	±	12.12
	15′	59.70	±	8.95	5.834	±	0.875	0.628	±	0.094	14.31	±	2.15	0.4574	±	0.0915	83.85	±	12.58
	20′	72.51	±	10.88	6.376	±	0.956	1.409	±	0.211	19.76	±	2.96	0.5105	±	0.1021	77.00	±	11.55
	25′	85.95	±	12.90	8.068	±	1.210	1.206	±	0.181	24.71	±	3.71	0.5845	±	0.1169	92.20	±	13.83
	30′	92.80	±	13.90	6.084	±	0.913	1.529	±	0.229	25.69	±	3.85	0.6295	±	0.1259	93.15	±	13.97
	35′	106.70	±	16.00	6.592	±	0.989	2.212	±	0.332	30.84	±	4.63	0.6320	±	0.1264	103.20	±	15.50
	40′	110.20	±	16.50	8.727	±	1.309	1.835	±	0.275	31.27	±	4.69	0.6945	±	0.1389	116.40	±	17.50
	45′	104.80	±	15.70	20.630	±	3.100	2.623	±	0.393	42.05	±	6.31	1.0130	±	0.2030	135.50	±	20.30
(1 g sample + 20 mL water)																			
45 S 5	5′	3.005	±	0.451	32.32	±	4.85	0.2654	±	0.0398	17.33	±	2.60	0.759	±	0.151	5.465	±	0.820
	10′	3.500	±	0.525	38.03	±	5.70	0.2513	±	0.0377	20.38	±	3.06	1.151	±	0.230	8.720	±	1.308
	15′	4.846	±	0.727	52.47	±	7.87	0.2975	±	0.0446	27.13	±	4.07	1.318	±	0.264	9.405	±	1.411
	20′	4.674	±	0.701	54.05	±	8.11	0.3171	±	0.0476	27.87	±	4.18	0.992	±	0.198	7.365	±	1.105
	25′	6.554	±	0.983	64.28	±	9.64	0.4870	±	0.0730	33.25	±	4.99	0.933	±	0.187	9.205	±	1.381
	30′	7.346	±	1.102	69.34	±	10.40	0.5298	±	0.0795	35.91	±	5.39	1.389	±	0.278	8.570	±	1.286
	35′	7.804	±	1.171	73.59	±	11.04	0.5471	±	0.0821	37.05	±	5.56	1.380	±	0.276	8.640	±	1.296
	40′	6.923	±	1.038	81.47	±	12.22	0.4400	±	0.0660	41.81	±	6.27	1.381	±	0.276	9.190	±	1.379
	45′	9.429	±	1.414	89.41	±	13.41	0.7341	±	0.1101	45.15	±	6.77	1.459	±	0.292	8.410	±	1.262
(1 g sample + 20 mL water)																			
BIOMIN F	5′	1.318	±	0.198	31.51	±	4.73	0.2407	±	0.0361	16.07	±	2.41	1.222	±	0.244	8.930	±	1.340
	10′	1.646	±	0.247	40.71	±	6.11	0.3540	±	0.0531	20.97	±	3.15	1.147	±	0.229	8.625	±	1.294
	15′	2.640	±	0.396	49.71	±	7.46	0.5373	±	0.0806	29.95	±	4.49	2.158	±	0.432	6.590	±	0.989
	20′	3.214	±	0.482	58.42	±	8.76	0.6159	±	0.0924	35.28	±	5.29	2.646	±	0.529	8.055	±	1.208
	25′	3.637	±	0.546	63.13	±	9.47	0.7395	±	0.1109	39.38	±	5.91	2.654	±	0.531	9.685	±	1.453
	30′	4.510	±	0.676	68.75	±	10.31	1.0600	±	0.1590	40.64	±	6.10	3.034	±	0.607	7.960	±	1.194
	35′	3.415	±	0.512	68.94	±	10.34	0.6251	±	0.0938	38.42	±	5.76	2.949	±	0.590	6.785	±	1.018
	40′	3.890	±	0.584	74.63	±	11.19	0.7593	±	0.1139	43.60	±	6.54	2.818	±	0.564	7.460	±	1.119
	45′	3.644	±	0.547	76.30	±	11.44	0.7536	±	0.1130	42.38	±	636	3.256	±	0.651	5.965	±	0.895
(1 g sample + 20 mL water)																			
S 53 P4	5′	1.927	±	0.289	21.80	±	3.27	0.1810	±	0.0272	12.26	±	1.84	0.395	±	0.079	3.910	±	0.587
	10′	2.732	±	0.410	30.16	±	4.52	0.5770	±	0.0866	15.79	±	2.37	0.772	±	0.154	3.761	±	0.564
	15′	1.520	±	0.228	34.16	±	5.12	0.2283	±	0.0342	18.07	±	2.71	1.089	±	0.218	3.651	±	0.548
	20′	4.038	±	0.606	47.57	±	7.14	0.3601	±	0.0540	24.79	±	3.72	1.546	±	0.309	4.224	±	0.634
	25′	4.370	±	0.655	51.11	±	7.67	0.4322	±	0.0648	29.46	±	4.42	1.959	±	0.392	4.532	±	0.680
	30′	3.698	±	0.555	47.32	±	7.10	1.0910	±	0.1640	26.30	±	3.94	2.022	±	0.404	4.257	±	0.638
	35′	4.863	±	0.729	55.40	±	8.31	0.5264	±	0.0790	31.32	±	4.70	2.079	±	0.416	3.715	±	0.557
	40′	5.390	±	0.808	60.84	±	9.13	0.4825	±	0.0724	34.99	±	5.25	2.167	±	0.433	3.266	±	0.490
	45′	4.920	±	0.738	65.30	±	9.79	0.3839	±	0.0576	35.69	±	5.35	2.396	±	0.479	3.082	±	0.462

## Data Availability

The data presented in this study are available on request from the corresponding author.

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
