# Peer review of "Investigating Bioactive-Glass-Infused Gels for Enamel Remineralization: An In Vitro Study"

_jfb, 2024, doi:10.3390/jfb15050119_

Round 1
Reviewer 1 Report
Comments and Suggestions for Authors
I think it's a new idea to turn an inorganic material into a gel.
However, the tables and graphs are difficult to read.
I think it would be easier to understand if the photographs in Figures 1 and 2 were made into schematic diagrams.
The words in the explanations of the graphs in Figures 4 and 5 are small and difficult to read for readers.
I personally think it would be better to use the table in Figure 7 as a graph.
Author Response
Dear Professor
Thank you very much for your time and comments regarding the text. We will try to respond to and correct all of them in the text.
- I think it's a new idea to turn an inorganic material into a gel.
Thank you for your comment. As we wrote in the introduction section, bioactive glasses have a high potential for releasing various ions that may be responsible for the process of enamel remineralization or dentinal tubule occlusion. Therefore, we wanted to test this type of materials and determine their properties if they are in the form of gels.
- However, the tables and graphs are difficult to read.
Thank you for your attention. It is extremely difficult to arrange graphics of the appropriate quality into a ready format for magazines. Additionally, changes to the graphics may be perceived as interference with the study. Below, the professor will find larger pictures with a rheometer test. When submitting the text for final publication, I will include these graphics.
- I think it would be easier to understand if the photographs in Figures 1 and 2 were made into schematic diagrams.
Thank you for your comment. Sometimes it's good to take photos to show what the test looked like. To dispel any doubts, we will also add schematic diagrams.
- The words in the explanations of the graphs in Figures 4 and 5 are small and difficult to read for readers.
Thank you for your attention. Better graphics quality was presented as an answer to point 2.
- I personally think it would be better to use the table in Figure 7 as a graph..
Thanks for comments. We thought about it for a while, but then one table would turn into 6 graphs. Some elements are released in amounts of 20-40 milligrams and others in trace amounts of 0.1-0.4 mg. Then the charts could be unreadable due to the scale.
Thank you for your comments and good luck in your further research!
authors
Reviewer 2 Report
Comments and Suggestions for Authors
Dear authors,
Thank you for giving me a chance to review your manuscript entitled " Investigating Bioactive Glass-Infused Gels for Enamel Remineralization: An In Vitro Study ".
Kindly find my comments below which may help you.
1. Table 2 to Table 6: As a result of the post hoc test, between which values did you find significant differences? The comparison target is unclear in this table. Correct, please.
2. Also, please change the commas to dots for the values in the table.
3. Please unify the values after the decimal point.
4. Please write references according to the submission guidelines.
Author Response
Dear Professor
Thank you very much for your time and comments regarding the text. We will try to respond to and correct all of them in the text.
- Table 2 to Table 6: As a result of the post hoc test, between which values did you find significant differences? The comparison target is unclear in this table. Correct, please.
These changes are counted from the point T=0 when the gels were made. It was corrected, thank you very much for your help.
- Also, please change the commas to dots for the values in the table.
It was corrected, thank you very much for your help (blue)
- Please unify the values after the decimal point.
It was corrected, thank you very much for your help. (blue)
- Please write references according to the submission guidelines.
It was corrected, thank you very much for your help.
Good lick in your further research!
Reviewer 3 Report
Comments and Suggestions for Authors
Review comments on Investigating Bioactive Glass-Infused Gels for Enamel Remineralization: An In Vitro Study
1. Lines 44-45 have line breaks in sentences; decimal point errors in all tables.
2. The content unit to be marked in Table 1 is mol% or wt%?
3. Figures 4 and 5 narrow the range of ordinates appropriately, and the font of the legend on the right is too small.
4. Line 68 “The unique remineralizing properties of bioactive glass have been acknowledged in dental research” supplements relevant references; line 75 “For this reason, different products have different mineralization effects” supplements relevant references.
5. Line 92 “Fluoride itself has bacteriostatic properties, which inhibits the demineralization process” needs to add relevant explanation.
6. The advantages of bioactive glasses in the 142-147 lines are not suitable in Materials Preparation.
7. Lines 424-428 are exactly the same as lines 462-465; lines 433-440 are exactly the same as lines 470-477.
8. The figures and labels in the text are not clear and appear confused.
Comments on the Quality of English LanguageMinor editing of English language required
Author Response
Dear Professor
Thank you very much for your time and comments regarding the text. We will try to respond to and correct all of them in the text.
- 1. Lines 44-45 have line breaks in sentences; decimal point errors in all tables.
Thank you for comments, It has been corrected.
- The content unit to be marked in Table 1 is mol% or wt%?
Thank you for comments, It has been corrected Values in [%] by weight
- 3. Figures 4 and 5 narrow the range of ordinates appropriately, and the font of the legend on the right is too small.
Thank you for your attention. It is extremely difficult to arrange graphics of the appropriate quality into a ready format for magazines. Additionally, changes to the graphics may be perceived as interference with the study. When submitting the text for final publication, I will include these graphics.
- Line 68 “The unique remineralizing properties of bioactive glass have been acknowledged in dental research” supplements relevant references; line 75 “For this reason, different products have different mineralization effects” supplements relevant references.
Thank you for your comment, the reference has been added
- Line 92 “Fluoride itself has bacteriostatic properties, which inhibits the demineralization process” needs to add relevant explanation.
Thank you for your comment, the reference has been added
Fluorine ions show the greatest activity towards bacteria at acidic pH, at a very low concentration of 0.1 mM it can completely stop glycolysis in intact Streptococcus mutans cells. The anti-caries effect of fluoride is complex and includes the effect on both bacteria and the formation of fluorohydroxyapatite in the mineral phases of mineral enamel. This compound has much lower solubility than hydroxyapatite.
- The advantages of bioactive glasses in the 142-147 lines are not suitable in Materials Preparation.
Thank you, this fragment has been removed.
- Lines 424-428 are exactly the same as lines 462-465; lines 433-440 are exactly the same as lines 470-477.
Repeated sentences have been removed. Thank you very much for your help
- The figures and labels in the text are not clear and appear confused.
Thank you for your help, to better illustrate the idea of the experiments we have added a schema to Figures 1 and Figure 2 to make this test more transparent
Good luck in your further research!
Round 2
Reviewer 3 Report
Comments and Suggestions for Authors
The authors responded to my comments . I recommend the publication.
Author Response
Dear Professor
thank you again very much for your time, and help in this manuscript.
have a nice day
authors